# Parasitic Helminths and Freshwater Fish Introduction in Europe: A Systematic Review of Dynamic Interactions

**Anaïs Esposito *** , **Joséphine Foata and Yann Quilichini ***

UMR (Unité Mixte de Recherche) 6134 CNRS (Centre National de la Recherche Scientifique)-Université de Corse Pascal Paoli Sciences Pour l'Environnement, 20250 Corte, France
* Correspondence: esposito_a@univ-corse.fr (A.E.); quilichini_y@univ-corse.fr (Y.Q.)

**Abstract:** The introduction of non-native freshwater fish is a primary cause of aquatic biodiversity loss at global scale. Such introductions have a severe impact on freshwater ecosystems in terms of competition, predation, habitat alteration, genetic pollution and transmission of diseases and parasites. A systematic review was conducted on the helminths parasites of freshwater fish in the context of species introduction and a total of 199 publications were retrieved between 1969 and November 2022. Several scenarios may arise when a new fish species arrive in a recipient area. Non-native fish hosts can co-introduce their parasites without transmitting them to native fish (e.g., the case of North American Centrarchidae and their Monogenea parasites). Another possible outcome is the transfer of these parasites to the native fish fauna (spillover, e.g., the cases of the Nematoda *Anguillicola crassus* Kuwahara, Niimi & Itagaki, 1974 and the Monogenea *Gyrodactylus salaris* Malmberg, 1957). Reciprocally, non-native fish hosts may acquire parasites in their new distribution range whether these parasites are native or were previously introduced (e.g., the cases of Ponto-Caspian Gobiidae and the Chinese sleeper *Perccottus glenii* Dybowski, 1877). Acquired parasites can then be spilled back to the native fauna. This phenomenon is of particular interest when non-native fish hosts influence the dynamics of zoonotic parasites.

**Keywords:** parasite co-introduction; spillover; spillback; *Anguillicola crassus*; Ponto-Caspian Gobiidae; North American Centrarchidae

**Key Contribution:** The interactions between native and non-native freshwater fishes and parasites were reviewed at European scale. Non-native fish tend to facilitate the spread of native and previously introduced parasites through a spillback effect and more attention should be brought to zoonosis in this context. Parasites infecting their fish host at an immature stage seem more successful in using non-native fish hosts.

## 1. Introduction

The introduction of non-native freshwater fish is one of the main causes of the decline of aquatic fauna at a global scale [1,2]. Species introduction can occur at three spatial scales: inter-continental, intra-continental and small/intra-country scale [3]. Fish introductions have occurred intentionally, for aquaculture, recreational fishing, ornamental, conservation or biological control purposes, or accidentally, e.g., through transport by ballast water or indirectly with the construction of inter-basin canals [3–5]. As an example, several freshwater fishes from North America were introduced in Poland between the 1860s and the end of the 1950s, of which the brown bullhead *Ameiurus nebulosus* (Lesueur, 1919), the chinook salmon *Oncorhynchus tshawytscha* (Walbaum, 1792) and the brook trout *Salvelinus fontinalis* (Mitchill, 1814) were able to maintain natural reproduction in the country [6]. In South Africa, the Mozambique tilapia *Oreochromis mossambicus* (Peters, 1852) was introduced for aquaculture in 1936 and is now established and widespread, including in estuarine and brackish environments [4]. The introduction of the mosquitofish *Gambusia affinis*

(Baird & Girard, 1853) in the Mediterranean island of Corsica at the end of the 19th century is a case of introduction for the purpose of biological control as part of an extensive malaria control program [7]. Consequences of freshwater fish introductions are various: alteration of interactions between fish (increase in prey availability for native predators, competition for trophic resource, increased predation pressure), alteration of habitat through changes in the trophic web structure, ecosystem function (trophic cascade) and eutrophication, genetic impact through hybridization and introgression, and socio-economic impact with financial loss for aquaculture and/or fisheries [4–6,8]. An additional impact of fish introductions is the co-introduction of associated parasites and diseases [4,5,8–10]. Non-native pathogens (viruses, bacteria, fungi and animal parasites) are frequently introduced along with their hosts in a recipient area [10,11]. The study of animal parasite carried by non-native hosts is of great importance as they may infect native populations of host species [12]. Helminths in particular encompass a vast diversity of organisms at different taxonomic levels: Annelida, Acanthocephala, Platyhelminthes, Cestodes, Trematodes, Nematodes, Pentastomida and Leeches [13]. Some of them are among the best-known examples of parasite transfer between non-native and native populations of host species, e.g., the Asian tapeworm *Schyzocotyle acheilognathi* (Yamaguti, 1934) is a known, important fish pathogen that spread in Australian freshwaters along with its original host the common carp *Cyprinus carpio* Linnaeus, 1758, and was transferred to several native fish species [14].

In Europe, the first recorded freshwater fish introduction is the translocation of the common carp *C. carpio* during the Roman era [15]. Large-scale introductions of freshwater fish have, however, been continuous since the early 19th century, with the introduction of North American Centrarchidae *Lepomis gibbosus* (Linnaeus, 1758) and *Micropterus salmoides* (Lacepède, 1802) and Salmonidae *Oncorhynchus mykiss* (Walbaum, 1792) and *Salvelinus* spp., and have shown a steady increase since the 1850s [16]. The estimates for the number of introduced freshwater fish in Europe range from 76 species when excluding intra-European introductions and 134 species or subspecies when including intra-European introductions [16,17]. The majority of introduced species are Cyprinidae and Salmonidae, with also numerically notable Cichlidae, Centrarchidae, Poeciliidae, Ictaluridae and Catostomidae, mainly originating for North-America and Asia [16].

The arrival of non-native host species in a recipient area can have several outcomes, parasitologically speaking (Figure 1):

(1) Parasite loss during translocation of their host: this can occur through two mechanisms, either by "missing the boat" (Figure 1a.1), when the introduced hosts do not carry the parasite, or by "drowning on arrival", when the host or the parasite fails to establish in the novel habitat (early extinction following host establishment, lack of suitable intermediate and/or final hosts in the recipient area) (Figure 1a.2) [18,19]. These mechanisms contribute to the often diminished parasite diversity observed in non-native organisms, e.g., [20,21]. This "release from the enemy" is often cited as a key factor in the success of non-native hosts in their new range [19,22,23]. The discretion of this specific case makes it rather difficult to monitor in the wild.

(2) Co-introduction of parasites with their host without transmission to native fauna (Figure 1b): Co-introduced parasites can establish and spread in their new range by only infecting their original host and not switching to native hosts. This absence of host switch can be attributed to a lack of suitable hosts in the recipient area and the parasite's host specificity [24].

(3) Transmission of novel parasites to native host (Figure 1c): Non-native hosts carrying parasites can transmit their parasites to native hosts, a host-switching mechanism termed 'parasite spillover' [12,24,25]. Pathogen co-introductions may give rise to particularly severe consequences because of the lack of co-evolution between non-native parasites and a native host, which thus does not possess adequate immune response to the infection [26].

(4) Parasite acquisition by non-native fish host: Depending on the host specificity of parasites already present in the recipient area (either native or previously introduced),

a non-native host may acquire new parasites in their introduced range [12]. These new interactions can result in parasite spillback, when a non-native species is a competent host for a native parasite and the presence of this additional host results in an increased opportunity to impact native hosts (Figure 1d.1) [27]. Non-native species can also act as sink hosts by being less suitable for native parasites, and thus reduce transmission to native hosts through a dilution effect (Figure 1d.2) [23,28]. The case where a newly acquired parasite can bear noticeable pathogenicity on a non-native should also be noted, e.g., [29], a case which is termed "suppressive spillover" by Chalkowsky et al. [30].

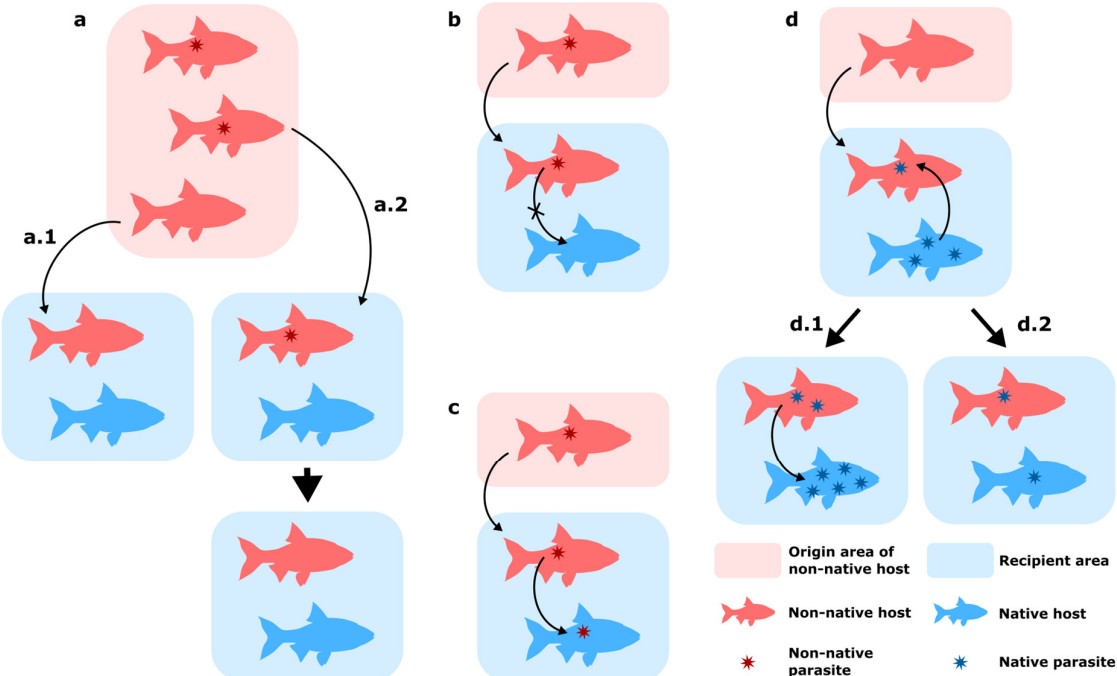

**Figure 1.** Possible outcomes for host–parasite systems after the introduction in a new habitat. (**a**) Parasite loss by either (**a.1**) missing the boat or (**a.2**) drowning on arrival (**b**), co-introduction of parasites with their host without transmission to native fauna, (**c**) transmission of novel parasites to native host (spillover) and (**d**) parasite acquisition by non-native fish host resulting in either (**d.1**) spillback effect or (**d.2**) dilution effect.

As with any other species introduction, freshwater fish non-native to Europe constitute host–parasite systems subject to the previously described mechanisms. The aim of this review was to provide a wide-ranging view of the parasitological outcomes of freshwater fish introductions in Europe, and illustrate them with well-studied cases involving helminths parasites. Our main focuses are as follows: to explore the temporal and geographical tendencies of the most thoroughly studied cases; identify trends associated with biotic parameters linked to the parasite or the host; isolate records of impacts associated with parasites in the context of species introduction; and highlight cases involving zoonotic parasites and potential human health-associated risks.

## 2. Material and Methods

A literature review was undertaken focusing on introductions of fish and parasitic helminths at a European scale. PRISMA guidelines [31] were followed to identify and select relevant studies. Because the registration in Prospero is not suitable for this manuscript, this review was not registered. Data gathered from these studies were compiled and analyzed. Articles taken into consideration were published in international journal indexed by Web of Science, Scopus and/or PubMed (Figure 2). Only articles published in English were

included in the review. The search date range was not limited by a starting date and ran to 2 November 2022, the date on which the search was carried out. The keywords used were Fish* AND Introduc* AND each term designating helminth parasites (Helminth* OR Platyhelminth* OR Trematod* OR Hirudinea OR Leech* OR Acanthocephal* OR Cestod* OR Nematod* OR Pentastomid* OR Digenea* OR Monogenea* OR Aspidogastrea*) with the 'topic' option to retrieve articles in which search terms appeared in the title, abstract and keywords. Only research articles were considered; literature reviews were excluded except when they also presented new data, in which case only the new data were compiled. Duplicates and inaccessible articles were excluded. Titles and abstracts were reviewed and the following inclusion/exclusion criteria were applied to the 1044 selected articles.

1. Did the study include a European country? Yes/No
2. Did the study include a fish host? Yes/No
3. Did the study include an introduced parasite helminth or a parasite helminth of an introduced fish host? Yes/No
4. Did the study include a freshwater environment? Yes/No
5. Were the studied fish wild-caught and the infestations natural? Yes/No

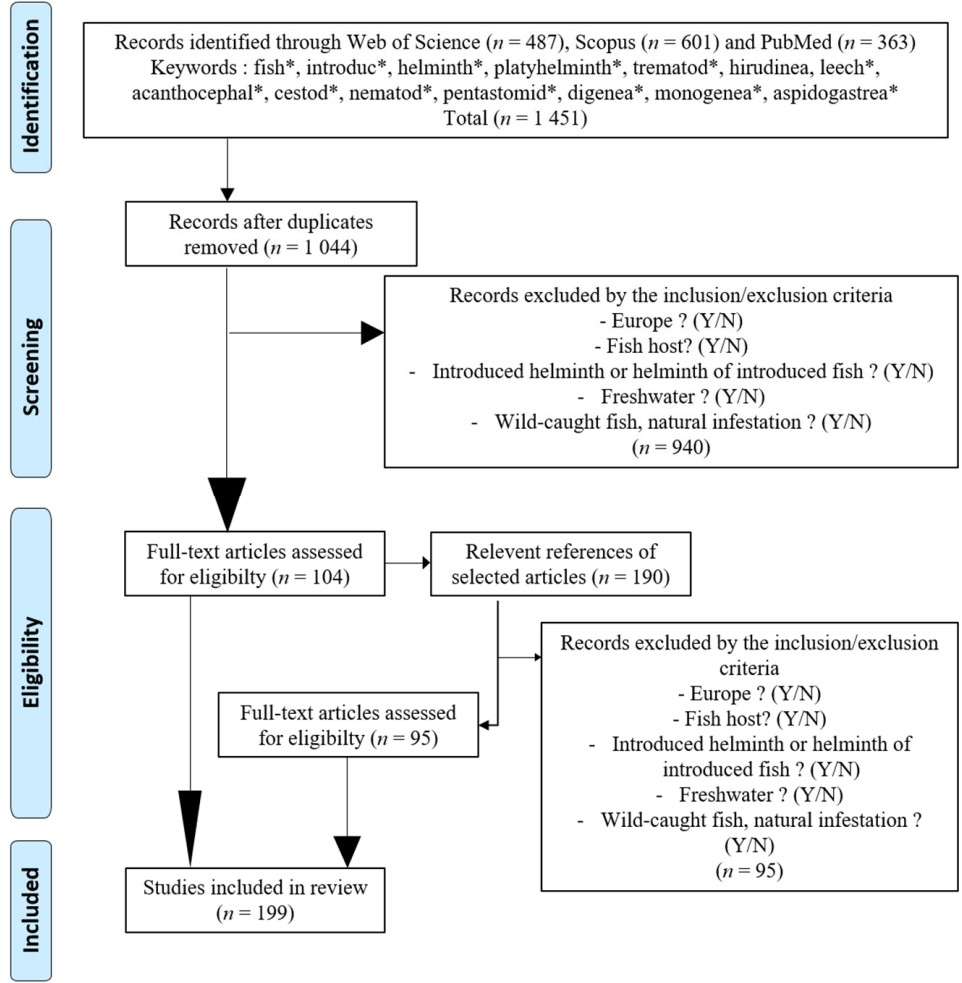

**Figure 2.** Workflow diagram used for the bibliographical search as defined in the PRISMA 2009 methodology, according to Moher et al. (2015). The '*' in search terms represent the wildcard symbol used to broaden the search to words starting with the same letters.

The European frontiers considered were biogeographical frontiers: the Ural and Caucasus mountain ranges and the Bosporus strait. Some articles included freshwater and brackish and/or marine environments, in which case only data concerning freshwater environments were taken into account. Likewise, when a study included results from both

natural and experimental infestations, only natural infestations were included. The articles were retained if the answer to all five question was 'Yes' and were discarded otherwise. The data of interest retrieved from the full texts were compiled into a database; articles not meeting the criteria were excluded and the bibliography of selected articles was reviewed. The inclusion/exclusion criteria previously described were applied to the new articles of interest from the bibliography and relevant studies were added to the review.

The information of interest with regard to the topic was compiled in an Excel matrix that was previously tested using a subsample of 20 articles. All articles were screened by the same one reviewer. The following information was collected:

1.  Main information concerning the study: title, date, authors.
2.  Location: country, watershed, number of sampling sites, habitat type (e.g., river, lake, reservoir), and island/mainland situation.
3.  Parasite-related information: taxonomy (phylum, class, subclass, order and species), status (native/non-native, and native distribution range for non-native), life cycle (direct/indirect and found on intermediate/final hosts), host specificity, microhabitat, zoonotic status, impact on fish host, socio-economic and/or ecological impact, both shown by the considered study and/or reported with the bibliography.
4.  Fish host-related information: taxonomy (family, species), status (native/non-native, and native distribution range for non-native), IUCN status, habitat type (e.g., demersal, benthopelagic) and diet, as retrieved from the Fishbase database (https://www.fishbase.se (accessed on 19 June 2023, [32]), and number of hosts examined.
5.  Methodology: method of detection and identification of parasites.

One line of the matrix corresponded to one parasite/host interaction, which is one record of one parasite in one host in one study. Only the native parasite/non-native host, non-native parasite/native host and non-native parasite/non-native hosts were taken into account. Native parasite/native host interactions were not compiled. Each of the synthesis presented in this review were illustrated by the most thoroughly studied (i.e., where the most articles were available) and relevant cases, and thus not all retrieved publications were cited in the present article.

## 3. Bibliographical Analysis

The bibliography review yielded a total of 199 articles and 969 parasite/host interactions. The first article retrieved was published in 1969 and was the only article selected for that year. During the first two decades, few papers were published. The number of publications started to rise in 1986, which can be explained mainly by the introduction of *Anguillicola crassus* Kuwahara, Niimi & Itagaki, 1974, in Europe and the growing interest of researchers in this pathogenic parasite of eel. The number of publications continued to increase and reached its maximum in 2011, with articles focused on a diversity of non-native parasites and parasites of non-native fish, e.g., parasite communities of *L. gibbosus* [33,34] and *Perccottus glenii* Dybowski, 1877 [35,36], the native *Pseudocapillaria tomentosa* (Dujardin, 1843) on the invasive *Pseudorasbora parva* (Temminck & Schlegel, 1846) [37] or *S. acheilognathi* and *Atractolytocestus huronensis* Anthony, 1958, both non-native Cestoda found on *C. carpio* [38,39]. Over the whole period (1969–2022), the mean number of articles was 3.7 per year, ranging from 0 to 13 articles.

Several non-native fish hosts and parasites attracted attention from researchers throughout Europe. The most widely studied non-native fish hosts were the Ponto-Caspian gobies (30 articles from 1994 to 2021), with the most focused-on species being *Neogobius melanostomus* (Pallas, 1814) (23 articles), followed by *Neogobius fluviatilis* (Pallas, 1814) (14 articles), *Ponticola kessleri* (Günther, 1861) (9 articles) and *Babka gymnotrachelus* (Kessler, 1857) and *Proterorhinus semilunaris* (Heckel, 1837) (7 articles each). The less studied *Knipowitschia caucasica* (Berg, 1916) and *Ponticola gorlap* (Iljin, 1949) were the focus of one article each [40,41]. Centrarchidae originating from North America, also drew attention from researchers: the pumpkinseed *L. gibbosus* was the focus of 28 articles and the largemouth black bass *M. salmoides* was studied four times. The Chinese sleeper *P. glenii*, an Odonbutidae originating from eastern Eurasia (Amur river drainage) was the focus of 25 articles. From the

host perspective, the family Gobiidae exhibited by far the greatest number of recorded host–parasite interactions (326 interactions), with all species being Ponto-Caspian invaders, followed by the Odontobutidae, exclusively represented by *P. glenii* (146 interactions), Centrarchidae (only *L. gibbosus* and *M. salmoides*, 110 interactions), Anguillidae (almost exclusively the native *Anguilla anguilla* (Linnaeus, 1758)), Cyprinidae (numerous species, 79 interactions) and Salmonidae (mainly native or stocked *Salmo trutta* Linnaeus, 1758, and stocked *O. mykiss*, 66 interactions). Seventy-nine articles focusing on or taking into account the invasive Nematoda *A. crassus* were retrieved, making this species the most studied non-native fish parasite in Europe. The Ponto-Caspian Acanthocephala *Pomphorhynchus laevis* (Zoega in Müller, 1776) Porta, 1908 was also focused on (12 articles), as well as the Asian Cestoda *Khawia japonensis* (Yamaguti, 1934) Hsü, 1935 and *Khawia sinensis* Hsü, 1935 (5 articles). In terms of retrieved interactions, the most represented parasites were Nematoda (32%) and Digenea (27%), followed by Monogenea (19%). Acanthocephala and Cestoda accounted for 12% and 10% of recorded interactions, respectively.

The retrieved articles included fish samples originating mainly from the United Kingdom (41 articles) and Germany (33 articles), with about half the articles focusing on *A. crassus*. They were followed by Poland and Hungary (17 articles each), Slovakia (14 articles), Norway (13 articles), France and Austria (12 articles each) and Czech Republic (10 articles). The limited number of articles retrieved from European Russia is biased by the limited accessibility to these articles and the fact that a large number of them were written in Russian.

The studied habitats encompassed a wide range of water body types, with the most focused on being rivers (104 articles), lakes (74 articles), reservoirs (23 articles) and ponds (20 articles). Some minor habitats included brooks, streams, channels, canals, side-arms and gravel pits. This wide diversity of water bodies represents so many potential recipient habitats for introduced fish and parasites, with various abiotic parameters and potential intermediate hosts and/or final hosts that may allow potential new parasites to complete their life cycle. Reservoirs in particular were shown to contribute to the biotic homogenization of fish hosts as native riverine fish tend to be replaced with cosmopolitan lentic species [42,43].

Freshwater fish parasites co-introductions occur at three distinct spatial scales: (1) inter-continental scale, when parasites are co-introduced into Europe from another continent, e.g., the Cestoda *Nippotaenia perccotti* (Akhmerov, 1941) and the Monogenea *Gyrodactylus perccotti* Ergens & Yukhimenko, 1973, introduced from eastern Asia along with their host *P. glenii* [44–47]; or the Ancyrocephalidae Monogenea introduced from North America with *L. gibbosus* and *M. salmoides* [33,48–50]; (2) intra-European scale, with parasites co-introduced from one region of Europe to another, e.g., the Monogenea *Dactylogyrus chondrostomi* Malevitskaia, 1941, *Dactylogyrus dirigerus* Gusev, 1966, *Dactylogyrus ergensi* Molnár, 1964 and *Dactylogyrus vistulae* Prost, 1957, probably arriving in France from eastern Europe with the Leuciscidae *Chondrostoma nasus* (Linnaeus, 1758) [51] or the Monogenea *Thaparocleidus vistulensis* (Sivak, 1932) Lim, 1996 introduced from Central Europe to Italy with *Silurus glanis* Linnaeus, 1758 [52–54]; and (3) small scale (intra-country scale), when fish are transferred from one waterbody to another within a relatively short distance, e.g., the Monogenea *Gyrodactylus aphyae* Malmberg, 1957, *Gyrodactylus macronychus* Malmberg, 1957, *Gyrodactylus magnificus* Malmberg, 1957 and *Gyrodactylus phoxini* Malmberg, 1957, probably introduced from one Norwegian watercourse to another [55], or the Cestoda *Dibothriocephalus ditremus* (Creplin, 1825), and *Dibothriocephalus dendriticus* (Nitzsch, 1824), probably spread from one subarctic Norwegian lake to another with *Gasterosteus aculeatus* Linnaeus, 1758 [56]. As already pointed out by Kuhn et al. [56], small-scale parasite co-introductions are the least studied scale while the research effort is mainly directed at inter-continental co-introductions.

Insular environments suffer from a lack of attention from parasitology research. Only one article considering one such habitat could be retrieved and concerned the Åland Islands in Finland [57]. The term 'island', in this context, refers to those with limited surface area (<30,000 km²). No article was retrieved concerning islands in the Mediterranean, islands being, however, particularly vulnerable to biological invasions [58,59].

## 4. Co-Introduction of Parasites with Their Fish Host without Transmission to Native Fish

### 4.1. The Case of North American Centrarchidae and Their Monogenea

North American Centrarchidae, i.e., pumpkinseed *L. gibbosus* and largemouth black bass *M. salmoides*, drew attention from researchers, with a total of 29 articles focusing on their parasites. Both species originate from eastern drainage systems of North America (Canada and USA). They were first introduced at the end of the 19th century to numerous countries in all regions of Europe to serve as gamefish and aquarium and garden pond fish [17]. Despite their relatively early introduction, research has only started to focus on these fishes' parasites since 1991–1995 with five articles studying *L. gibbosus*. They have attracted more interest recently, with most articles concerning parasites of both Centrarchidae published between 2016–2020 (Figure 3).

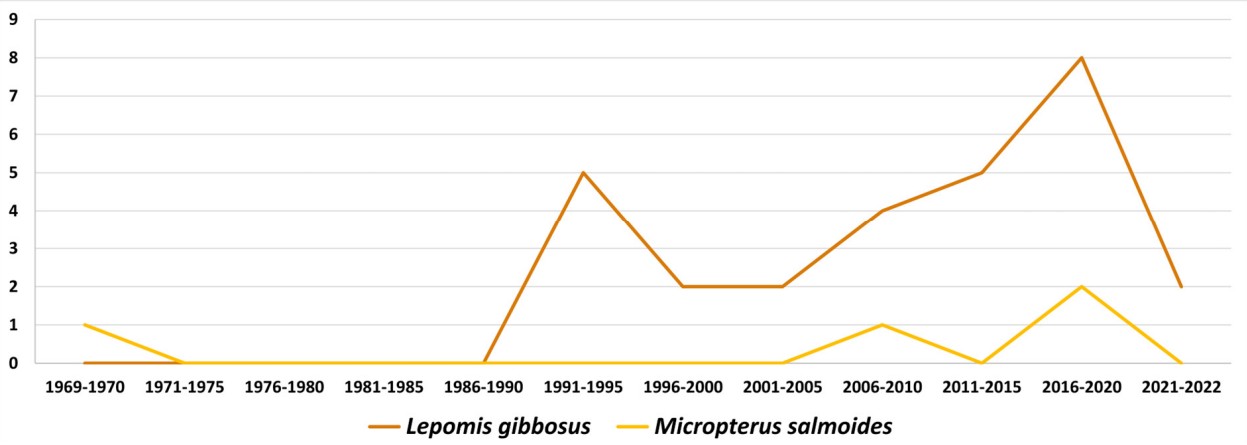

**Figure 3.** Progression over time of published studies concerning North American Centrarchidae in Europe.

Since its introduction, *L. gibbosus* spread successfully into adjacent water bodies and established populations throughout Europe [60]. *Micropterus salmoides* is now mainly found in the Iberian Peninsula, France and Italy [8], and its parasites were mostly studied in southern Europe. Researchers who showed interest in these species' parasites carried out their studies mostly in eastern Europe (16 articles, e.g., [48,61–64]), followed by western (e.g., [49,65–67]) and southern (e.g., [53,54,68]) Europe (Figure 4).

Of all the retrieved articles concerning parasites of North American Centrarchidae in Europe, 10 focused exclusively on Monogenea. At least nine species of North American Monogenea were co-introduced to Europe, with the majority being Centrarchidae-specific parasites belonging to the Ancyrocephalidae family. Three species were widely found parasitizing *L. gibbosus* in Europe, namely *Actinocleidus recurvatus* Mizelle & Donahue, 1944, *Onchocleidus dispar* Mueller, 1936, and *Onchocleidus similis* Mueller, 1936 (Table 1). In contrast, several species seem to have a restricted non-native range, e.g., *Cleidodiscus robustus* Mueller, 1934, *Onchocleidus acer* Muller, 1936, and *Gyrodactylus macrochiri* Hoffman & Putz, 1964, were only reported from France [20,21] and *Gyrodactylus avalonia* Hanek & Threlfall, 1969, only from Ukraine [61].

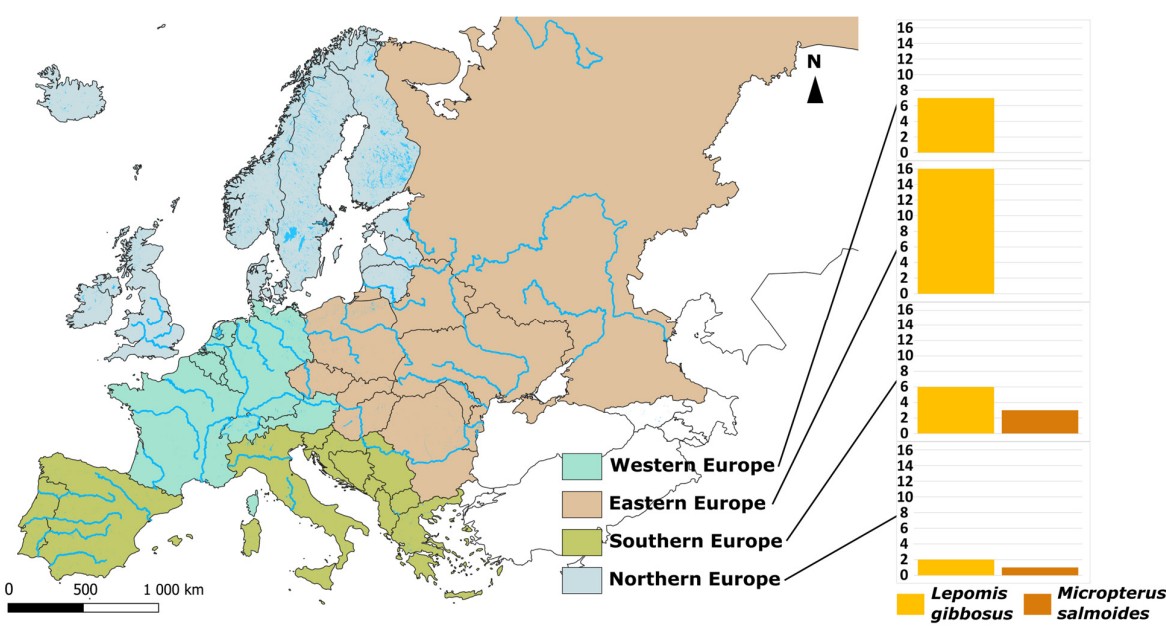

**Figure 4.** Geographical distribution of published studies concerning parasites of North American Centrarchidae in Europe.

**Table 1.** North American Monogenea co-introduced in Europe with their Centrarchidae hosts.

| Parasite | | Host Species | Locality |
|---|---|---|---|
| **Family** | **Species** | | |
| Ancyrocephalidae | *Actinocleidus oculatus* | *Lepomis gibbosus* | France [20,21], Germany [21,49], Italy [52–54] |
| | *Actinocleidus recurvatus* | *Lepomis gibbosus* | Austria [21], Croatia [33], France [20,21], Germany [21,49], Italy [52–54], Slovakia [33] |
| | *Actinocleidus* sp. | *Lepomis gibbosus* | Austria [21], France [21], Germany [21,49] |
| | *Cleidodiscus robustus* | *Lepomis gibbosus* | France [20,21] |
| | *Onchocleidus acer* | *Lepomis gibbosus* | France [20] |
| | *Onchocleidus dispar* | *Lepomis gibbosus* | Austria [21], Bulgaria [21,33,48], Croatia [33], Czech Republic [21,33], France [20,21], Germany [21,49], Italy [52–54], Portugal [69], Slovakia [33], Ukraine [63], United Kingdom [34] |
| | | *Micropterus salmoides* | Portugal [69] |
| | *Onchocleidus principalis* | *Lepomis gibbosus* | Portugal [69] |
| | | *Micropterus salmoides* | Italy [54], Portugal [69], United Kingdom [70] |
| | *Onchocleidus similis* | *Lepomis gibbosus* | Austria [21], Bulgaria [21,33,48], Croatia [33], Czech Republic [21,33], France [20,21], Norway [50], Germany [21,49], Italy [52–54], Slovakia [33], Ukraine [61] |
| | *Onchocleidus* sp. | *Lepomis gibbosus* | Germany [21,49], Norway [50] |
| | *Unidentified Ancyrocephalidae* | *Lepomis gibbosus* | Austria [21], France [21] |
| Gyrodactylidae | *Gyrodactylus avalonia* | *Lepomis gibbosus* | Ukraine [61] |
| | *Gyrodactylus macrochiri* | *Lepomis gibbosus* | France [20,21] |

None of the reviewed articles reported any adverse effect of the co-introduction of these parasites, either on their hosts or in the recipient area. No documented case of transfer of these Monogenea to the native fish fauna could be found and they thus appear harmless

for the time being. The only exception to this observation is the case of *G. avalonia*, as explained by Kvach et al. [61]; this species shows a low host specificity as well as a broad tolerance to environmental conditions and its principal host, *G. aculeatus*, is abundant in the Danube delta, where the parasite has been reported. For these reasons, the spread of *G. avalonia* is likely, and could be a threat as this Monogenea was shown to transfer pathogenic bacteria to its host [71].

### 4.2. Other Notable Co-Introductions of Fish Parasites in Europe

Other examples of parasite co-introduction with no transmission to native host are three specialists of the East Asian Chinese sleeper, *P. glenii*, an odontobutid introduced to central and eastern Europe as a result of aquaculture, release by an aquarist and their use as live bait by anglers [72,73]. One Cestoda, and *N. perccotti*, and one Monogenea, *G. perccotti* were introduced along with their host, and were recovered in Europe only from their host, e.g., [40,74]. As *N. perccotti* and *G. perccotti* are stenoxenous, the only impact they may have on native fish species is to give them a competitive advantage over the invasive *P. glenii*, especially since their abundance was qualified as relatively high in some invaded areas [75,76].

## 5. Co-Introduction of Parasites with Their Fish Host with Transmission to Native Fish

The infection of a new host species by a parasite endemic to one host species is a well-known mechanism in invasion biology called spillover [25,30]. The most striking and most studied case of this phenomenon in European fish is the infection of the European eel *Anguilla anguilla* by the Asian Nematoda *A. crassus*.

### 5.1. The Case of Anguillicola crassus

With 79 articles retrieved through the present bibliographical analysis, *A. crassus* is the parasite that has attracted the most attention from the scientific community. This Rhabditidae is thought to have been introduced in Germany in the 1980s with the importation of its native host *Anguilla japonica* Temminck & Schlegel, 1846, from Taiwan for aquaculture purposes.

The first article concerning *A. crassus* retrieved dates back to 1987 with the study of the then-recently introduced parasite by Taraschewki et al. [77] in Germany. Earlier occurrences were recorded but were either incorrectly identified or not available in English, so they were not included in the present study. The number of records rose during the following half-decade (1991–1995) to reach its maximum, and then started to decrease until the present day (Figure 5).

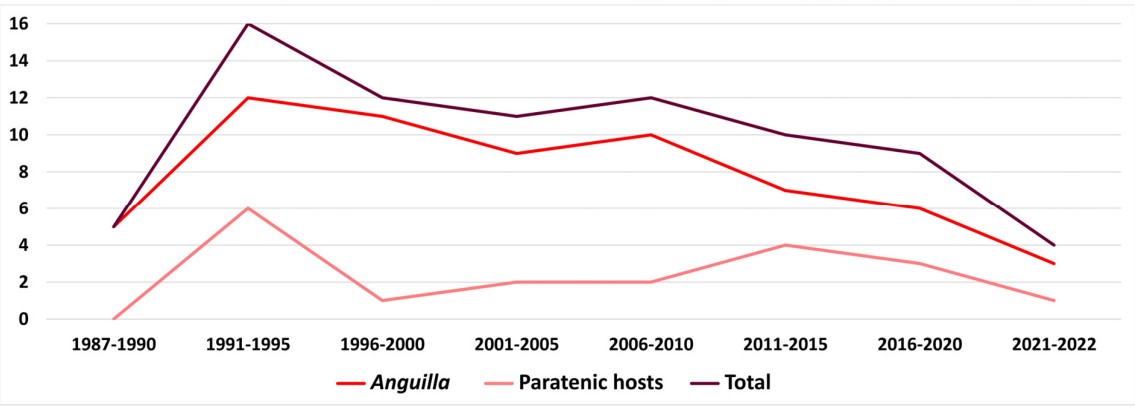

**Figure 5.** Progression over time of published studies concerning *A. crassus* in Europe.

The parasite then spread throughout the distribution range of its newly acquired host in Europe and North Africa [78–81]. *Anguillicola crassus* was extensively studied in western Europe (34 articles retrieved), eastern Europe (23 articles) and northern Europe (24 articles)

(Figure 6). It was a little less studied in southern Europe, but six articles were nonetheless retrieved. In western and eastern Europe, the attention was focused not only on the final host, but also on paratenic fish hosts (11 studies each).

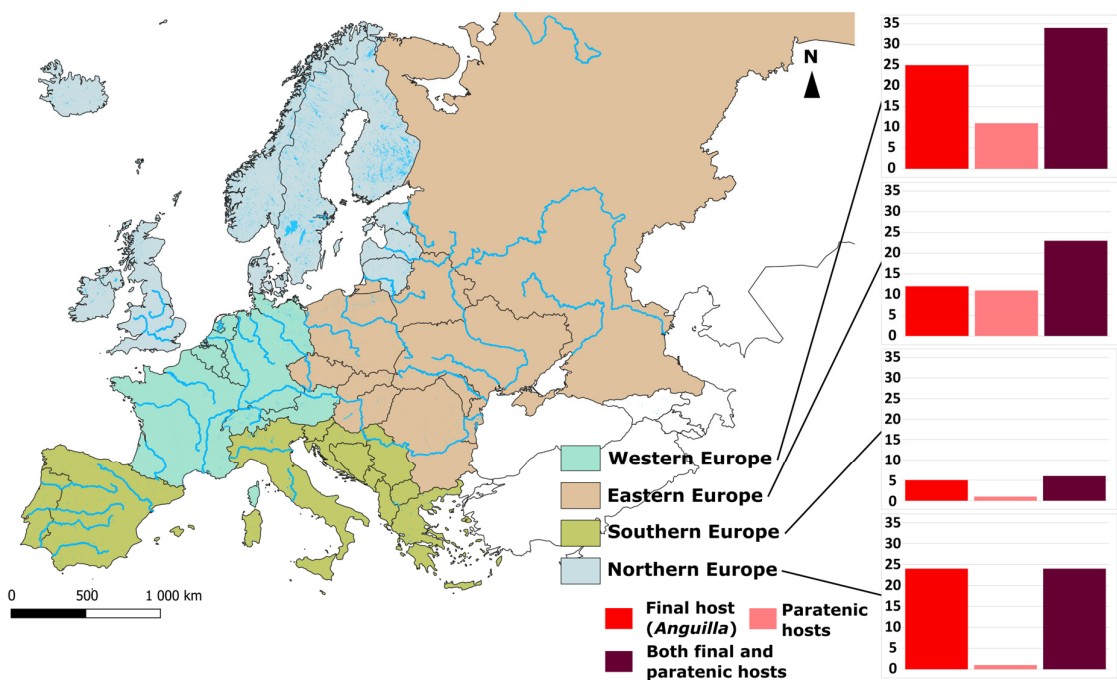

**Figure 6.** Geographical distribution of published studies concerning *Anguillicola crassus* in Europe.

### 5.1.1. Pathogenicity of *Anguillicola crassus*

*Anguillicola crassus* is known to be pathogenic to *A. anguilla* and is frequently cited as one of the threats involved in the decline of this critically endangered species [82], along with other anthropogenic and natural factors such as barriers to migration, climate and oceanic currents changes, loss and degradation of habitat, pollution, predation, legal and illegal exploitation and trade, non-parasitic invasive species, diseases, and parasites [83,84]. Numerous studies highlighted the physical and physiological impacts of *A. crassus* on *A. anguilla* at the individual level. Symptoms include numerous alterations and deformities of the swimbladder altering its integrity, such as thickening of the swimbladder wall and fibrosis [85–96], reduced elasticity due to scar tissue resulting from perforation [97], inflammation [87,95,98–100], hemorrhages or hemorrhagic ulcers [87,89,91,92,95,100], rupture [89], blood vessel dilation or congestion [92,98,100]. The infection by *A. crassus* can also result in physiological impacts, such as decreased stress tolerance [90,95,101], exacerbation of corticoid stress response associated with severe hypoxia, increased metabolic cost, increased plasma cortisol concentration and lack of hyperglycemic response [102]. An impairment of the defense capacity against reactive oxygen species was also noted [103]. At least one episode of mass mortality linked to *A. crassus* infection was recorded, in lake Balaton, Hungary [92]. Anguillicolosis was reported to impair the eels' swimming performances with a decreased cruising speed and increased energy consumption and an avoidance of accelerating flow [104,105]. Because of the damage to the functional capacity of an essential hydrostatic organ and the impact on swimming capacity, *A. crassus* is feared to hamper the transatlantic spawning migration of its host, thus contributing to its decline [104]. From a socio-economic perspective, *A. crassus* appears to be a threat to aquaculture production of eels by compromising their growth and yield [106].

### 5.1.2. Advances in Detection Methods

As *A. anguilla* is a critically endangered species, there is a need to carry on the study of its parasites, especially pathogenic ones, but there is also an urgent need to develop

ethical, non-lethal methods to reduce the impact of research on its populations [107]. For this reason, various methods were tested for the detection of *A. crassus*. Beregi et al. [91] showed the potential of a radiography approach to assess the infection in the swimbladder and the severity of pathological changes while avoiding harming the fish. However, Frisch et al. [108] came to the conclusion that computed radiography, computed tomography and magnetic resonance imaging were not sensitive enough to replace post-mortem examination, and were not suited to the diagnosis of mildly infected eels. Recently, a protocol of molecular detection in fecal samples using specific microsatellite markers yielded promising results while being easy to implement and non-lethal [107].

### 5.1.3. Factors Involved in *A. crassus* Success and Importance of Paratenic Hosts

As Kennedy and Fitch [109] demonstrated, the successful colonization of Europe by *A. crassus* can be explained by some of its characteristics: a high reproductive potential, a relatively simple life cycle and low intermediate host-specificity, a capacity of both eggs and stage 2 larvae to survive and remain infective for a long period in freshwater and up to two weeks in seawater, a capacity to infect any size of eel and to transfer from eel to eel, a widespread final host tolerant to diverse habitat conditions, and a capacity to survive and reproduce under any conditions withstood by its host. Its capacity to use several species of the genus *Anguilla* as a final host can also explain the further spread of this parasite to the American eel *Anguilla rostrata* (Lesueur, 1817), both in its native [110] and introduced range [111]. It is worth noting that the low intermediate host specificity also applies to paratenic hosts, as a lot of fish species were found to be suitable hosts for *A. crassus* larvae throughout Europe (Table 2). Indeed, of the retrieved articles, a total of 19 studies focused or took into account paratenic fish hosts. This capacity to use a wide range of paratenic hosts is undoubtedly an additional factor explaining the successful spread of this Nematoda in Europe, and its rapid range expansion [94,112–114]. *Anguillicola crassus* can use paratenic hosts pertaining to at least 12 families, and both native and non-native fish hosts. The most diversified native hosts are cyprinids, leuciscids and percids. A few examples of non-native paratenic hosts are the North American Centrarchidae *L. gibbosus* [94,112,115–117], the Central American cichlid *Amatitlania nigrofasciata* (Günther, 1867) [118], the Asian cyprinid *P. parva* [112,116,117] and a total of five well-studied Ponto-Caspian gobids [119–122]. The majority of these paratenic hosts are species living in close association with the substrate (benthopelagic and demersal species), as was previously pointed out in both Belgium and Sweden [94,123]. Most are known to consume zooplankton, which likely is their infection route as young *A. crassus* larvae use small cyclopoid copepods as intermediate hosts [124].

Invasional meltdown could also play a role in the spread of this parasite [125]. According to the invasional meltdown hypothesis, if several species invade the same habitat, then they can facilitate one another's establishment by acting as a food or energy resource for the other [126,127]. In the case of *A. crassus*, the Ponto-Caspian *N. melanostomus* introduced its Acanthocephala parasite *P. laevis* in the river Rhine since the 1990s [128], and cysts of *P. laevis* are suggested to serve as a 'hideout' allowing *A. crassus* to evade its paratenic host's immune response [129]. The presence of the co-introduced *P. laevis* could thus be another factor facilitating the spread of *A. crassus* through invasional meltdown [125,128,129].

**Table 2.** Paratenic hosts of *A. crassus* in Europe.

| Family | Species | Origin | Habitat | Diet | Locality |
|---|---|---|---|---|---|
| Centrarchidae | *Lepomis gibbosus* | Non-native | Benthopelagic | Nekton, Zoobenthos, Zooplankton, Detritus | Belgium [94], Czech Republic [21], Hungary [112,115–117] |
| Cichlidae | *Oreochromis niloticus* | Non-native | Benthopelagic | Zoobenthos, Zooplankton, Detritus, Plants | Belgium [94] |
| | *Amatitlania nigrofasciata* | Non-native | Benthopelagic | Nekton, Zoobenthos, Detritus | Germany [118] |
| Cyprinidae | *Alburnus alburnus* | Native | Benthopelagic | Zoobenthos, Zooplankton, Detritus, Plants | Belgium [94], Hungary [112,115–117] |

**Table 2.** *Cont.*

| Family | Species | Origin | Habitat | Diet | Locality |
|---|---|---|---|---|---|
| | *Blicca bjoerkna* | Native | Demersal | Zoobenthos, Zooplankton, Detritus, Plants | Hungary [112,116,117] |
| | *Carassius carassius* | Native | Benthopelagic | Nekton, Zoobenthos, Zooplankton, Detritus | Hungary [116,117] |
| | *Carassius gibelio* | Unclear | Benthopelagic | Zoobenthos, Detritus | Hungary [112,116,117] |
| | *Cyprinus carpio* | Native | Benthopelagic | Nekton, Zoobenthos, Zooplankton, Detritus, Plants | Hungary [112,115–117] |
| | *Leuciscus aspius* | Native | Benthopelagic | Zooplankton, Nekton | Hungary [112,116,117] |
| | *Phoxinus phoxinus* | Native | Demersal | Nekton, Zoobenthos, Zooplankton, Detritus, Plants | France [130] |
| | *Pseudorasbora parva* | Non-native | Benthopelagic | Nekton, Zoobenthos, Zooplankton, Plants | Hungary [112,115–117] |
| | *Rhodeus amarus* | Native | Benthopelagic | Zoobenthos, Plants | Hungary [112,116,117] |
| | *Romanogobio albipinnatus* | Native | Benthopelagic | Zoobenthos | Hungary [116,117] |
| | *Scardinius erythrophthalmus* | Native | Benthopelagic | Nekton, Zoobenthos, Zooplankton, Detritus, Plants | Belgium [94], Hungary [112,116,117] |
| | *Tinca tinca* | Native | Demersal | Nekton, Zoobenthos, Zooplankton, Detritus | Belgium [94], Hungary [112,116,117] |
| Esocidae | *Esox lucius* | Native | Pelagic | Nekton, Zoobenthos, Zooplankton | Hungary [112,116] |
| Gasterosteidae | *Gasterosteus aculeatus* | Native | Benthopelagic | Nekton, Zoobenthos, Zooplankton, Plants | Belgium [94], France [130] |
| | *Pungitius pungitius* | Native | Benthopelagic | Nekton, Zoobenthos, Zooplankton | France [130] |
| Gobiidae | *Babka gymnotrachelus* | Non-native | Benthopelagic | Nekton, Zoobenthos | Poland [119] |
| | *Neogobius fluviatilis* | Non-native | Benthopelagic | Nekton, Zoobenthos, Zooplankton | Germany [119,120], Hungary [112,116,117] |
| | *Neogobius melanostomus* | Non-native | Demersal | Zoobenthos | Austria [121,122,131], Croatia [122], Czech Republic [113], Germany [119,120,132], Slovakia [122] |
| | *Ponticola kessleri* | Non-native | Benthopelagic | Nekton, Zoobenthos, Zooplankton, Plants | Germany [119,120], Slovakia [131,133] |
| | *Proterorhinus semilunaris* | Non-native | Benthopelagic | No data available | Germany [119,120] |
| Gobionidae | *Gobio gobio* | Native | Benthopelagic | Zoobenthos, Zooplankton, Plants | Belgium [94], Hungary [112] |
| Ictaluridae | *Ameiurus nebulosus* | Non-native | Demersal | Nekton, Zoobenthos, Zooplankton, Plants | Belgium [94], Hungary [116] |
| Leuciscidae | *Abramis brama* | Native | Benthopelagic | Nekton, Zoobenthos, Zooplankton, Detritus, Plants | Hungary [112,115,116] |
| | *Chondrostoma nasus* | Native | Benthopelagic | Detritus, Plants | Belgium [94] |
| | *Leuciscus idus* | Native | Benthopelagic | Nekton, Zoobenthos, Zooplankton | Belgium [94] |
| | *Leuciscus leuciscus* | Native | Benthopelagic | Zoobenthos, Zooplankton, Detritus, Plants | Belgium [94] |
| | *Rutilus rutilus* | Native | Benthopelagic | Zoobenthos, Zooplankton, Detritus, Plants | Belgium [94], Hungary [112,115–117] |
| | *Squalius cephalus* | Native | Benthopelagic | Nekton, Zoobenthos, Zooplankton, Plants | Belgium [94] |
| Osmeridae | *Osmerus eperlanus* | Native | Pelagic-neritic | Nekton, Zoobenthos, Zooplankton | Netherlands [100] |
| Percidae | *Gymnocephalus cernua* | Native | Benthopelagic | Nekton, Zoobenthos, Zooplankton, Detritus, Plants | Belgium [94], Germany [114], Hungary [112,115–117], Poland [114], United Kingdom [134] |
| | *Perca fluviatilis* | Native | Demersal | Nekton, Zoobenthos, Zooplankton | Belgium [94], Hungary [112,116] |
| | *Sander lucioperca* | Native | Pelagic | Nekton, Zoobenthos, Zooplankton | Belgium [94], Hungary [112,115,116] |
| Siluridae | *Silurus glanis* | Native | Benthopelagic | Nekton, Zoobenthos, Zooplankton | Hungary [112,116,117] |

*Anguillicola crassus* larvae usually seem to cause little to no damage to its paratenic hosts as no pathological change nor demonstrable sign of host reaction could be shown in *Osmerus eperlanus* (Linnaeus, 1758) and *N. fluviatilis* [100,117]. Székely [117] noted the presence of proliferating tissue and connective tissue capsule in *Alburnus alburnus* (Linnaeus, 1758) and granulation tissue in *Gymnocephalus cernua* (Linnaeus, 1758), and parasite encapsulation was shown in a wide variety of hosts [112]. The only strong reactions were found in *S. glanis* with the presence of numerous nodules on the serous membrane around the gut and stomach, and in *Rutilus rutilus* (Linnaeus, 1758), which showed necrosis [117].

*5.2. The Case of Gyrodactylus salaris in Norway*

Co-introduced pathogenic parasites can also be external parasites with a direct cycle, as with the case of the Monogenea *Gyrodactylus salaris* Malmberg, 1957 in Norway. *Gyrodactylus salaris* was introduced to Norway in the 1970s through stocking of infected *Salmo salar* Linnaeus, 1758 smolts from infected hatcheries [135–137]. The arrival of this species in Norway has had serious consequences, both on the *S. salar* populations and from a socio-economic perspective. The most striking consequence was the heavy mortalities and quasi-disappearance of salmon parr populations in numerous rivers and lakes in Norway. This near-extinction of locally adapted stocks was accompanied by a serious and continuous risk of introducing the parasite from infected to neighboring rivers [135–139]. *Gyrodactylus salaris* caused secondary infections, e.g., *Saprolegnia* in its host, as well as osmoregulation imbalance due to its attachment and feeding on the skin and fins [139,140]. Additionally, this infection had a severe impact on local fishing tourism and led to costly countermeasures and substantial economic damages ($500,000,000 in Norway) [137,141]. The Norwegian authorities had to resort to rotenone treatments to eradicate the infected fish from the rivers [137]. An advantage that may have played a part in the success of *G. salaris* in Norway is its ability to use other Salmonidae as reservoir hosts, such as *Salvelinus alpinus* (Linnaeus, 1758) and *S. trutta* [136,142,143].

*5.3. Other Notable Co-Introductions of Parasites to Native Fish in Europe*

A less-known case of transmission of a pathogenic parasite from a non-native to a native fish host is the case of the transfer of the Digenea *Apatemon gracilis* (Rudolphi, 1819) from the invasive Cottidae *Cottus gobio* Linnaeus, 1758, to the native *Salmo salar*, in Finland, described by Ieshko et al. [144]. This parasite is capable of causing hemorrhages and can lead to mortality in juvenile fish in the event of a high infection intensity [144].

The transmission of novel parasites to native fish not only impacts the new host but can also have repercussions on native parasites through competition. A total of 12 articles focusing on *Pseudodactylogyrus anguillae* (Yin & Sproston, 1948) and/or *Pseudodactylogyrus bini* (Kikuchi, 1929) infections in *A. anguilla* were retrieved. *Pseudodactylogyrus* were first recorded in Europe in a Soviet Union eel farm before spreading throughout its new host's range [145]. The parasite is known to cause gill impairment in its host [146], but was also reported to impact local parasite communities as it competed with and led to the disappearance of the native eel Monogenea *Gyrodactylus anguillae* Ergens, 1960, in the United Kingdom [147]. This situation is an example of ecological impact and loss of native biodiversity due to a biological invasion.

In some cases, the newly acquired parasite does not seem to noticeably impact its new host or habitat, e.g., the North American Acanthocephala *Paratenuisentis ambiguus* (Van Cleave, 1921) was not reported to affect its new host *A. anguilla* [77,148,149].

**6. Non-Native Fish Host and Parasite Acquisition in the Recipient Area**

Introduced fish host species may act as a suitable host for parasite species already present in the recipient habitat, whether these parasites are native or originate from a previous introduction. It is accepted that parasites infecting non-native hosts in the recipient area are mainly generalists capable of infecting a wide range of species [150,151].

*6.1. Ponto-Caspian Gobiidae*

With 30 articles retrieved, Ponto-Caspian Gobiidae were the most focused on non-native fish taxon. The three first articles retrieved in which host–parasite interactions were noted for a Ponto-Caspian Gobiidae correspond to the rise in interest toward the invasive *A. crassus* and record *N. fluviatilis* as a paratenic host for this eel pathogen [112,116,117]. The invasion by Ponto-Caspian Gobiidae is considered one of the most impressive invasions of European continental waters in recent years [3]. Four species (*Proterorhinus marmoratus* (Pallas, 1814), *N. melanostomus*, *N. fluviatilis* and *P. gorlap*) began to expand up the river Volga during the period 1970–2000 [3]. The construction of canals connecting contiguous basins played a role in the spread of these species, e.g., *N. fluviatilis* and *B. gymnotrachelus* were able to expand their range up the river Dnieper via the Pripyat-Bug canal connecting the river Vistula and the river Dnieper [152]. The first range expansion was noted in the 1960s for *N. fluviatilis* in the river Danube in Serbia, then in Hungary in the late 1990s and in Slovakia in the early 2000s [153–155]. The interest shown toward these fishes' parasite communities started to increase shortly after, with the first retrieved article published in 2005 [133] (Figure 7). Of the four mainly studied Ponto-Caspian Gobiidae, the research effort directed at parasite communities peaked during the period 2011–2015 for *N. fluviatilis*, *N. melanostomus* and *P. semilunaris*. Research continues to be carried out on these species even though with seemingly reduced intensity over the past decade.

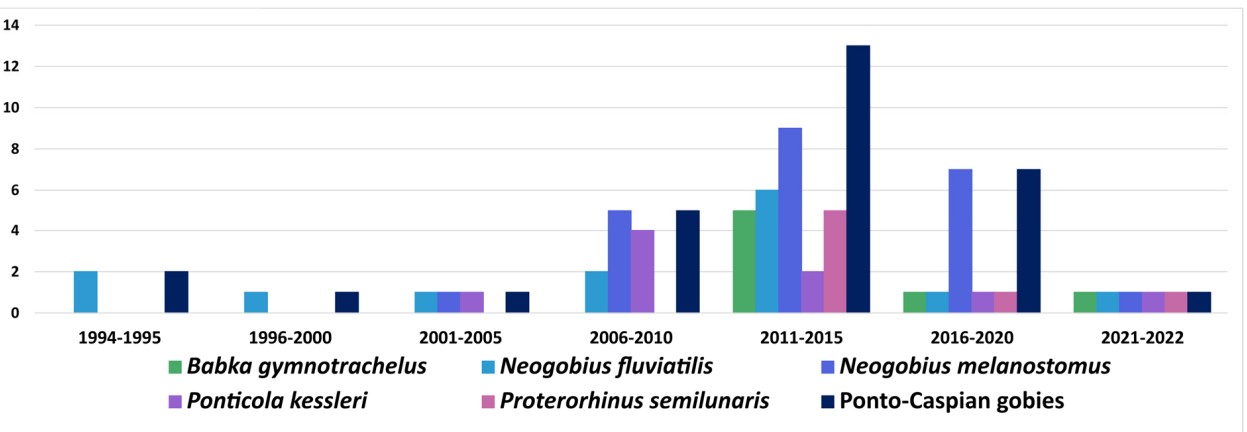

**Figure 7.** Progression over time of published studies concerning Ponto-Caspian Gobiidae in Europe.

Eastern and Western Europe were the most involved regions in the study of Ponto-Caspian Gobiidae parasites, with 19 and 16 articles, respectively. *N. fluviatilis* was more focused on in western Europe whereas *B. gymnotrachelus* and *N. fluviatilis* were more studied in eastern Europe, including in the lower Volga river [156]. Few articles were retrieved from southern Europe (two in total), directed only at *N. melanostomus* and *P. kessleri* (Figure 8). No publication was retrieved from northern Europe. This distribution pattern of publications is consistent with the invaded range of Ponto-Caspian Gobiidae.

6.1.1. Acquisition and Subsequent Spread of Native Parasite

According to the reviewed articles, at least 43 parasites species were recovered from Ponto-Caspian Gobiidae in their non-native range. Among these species, several were acquired by Gobiidae in their newly invaded habitat. A first example is the Digenea *Bucephalus polymorphus* von Baer, 1827, which was supposed to be introduced with Ponto-Caspian Gobiidae in Austria [157], but not in other regions, such as the river Morava basin, in which the parasite was known to occur since the 1950s [158]. As the occurrence of this parasite was not documented from the native area of Ponto-Caspian Gobiidae [121,122,131], it is likely that its acquisition occurred in their non-native range and that they subsequently played a role in its spread to Austria. *B. polymorphus* was recorded from the four principal species of Ponto-Caspian Gobiidae in several countries in western and eastern

Europe [119,121,122,159–161]. The hypothesis of an increase in parasite abundance by integration of these novel hosts has been put forward in several publications [119,160,162], as Mierzejewska et al. [160] noted a reinforcement of parasite population through infection of new hosts (*B. gymnotrachelus* and *N. fluviatilis*), and Ondračková et al. [162] reported an increase in prevalence and abundance of *B. polymorphus* metacercariae in both native and non-native fish hosts after the invasion of Czech Republic by Ponto-Caspian Gobiidae. It thus seems likely that non-native Gobiidae will facilitate the transmission and spread of this Digenea through a spillback effect.

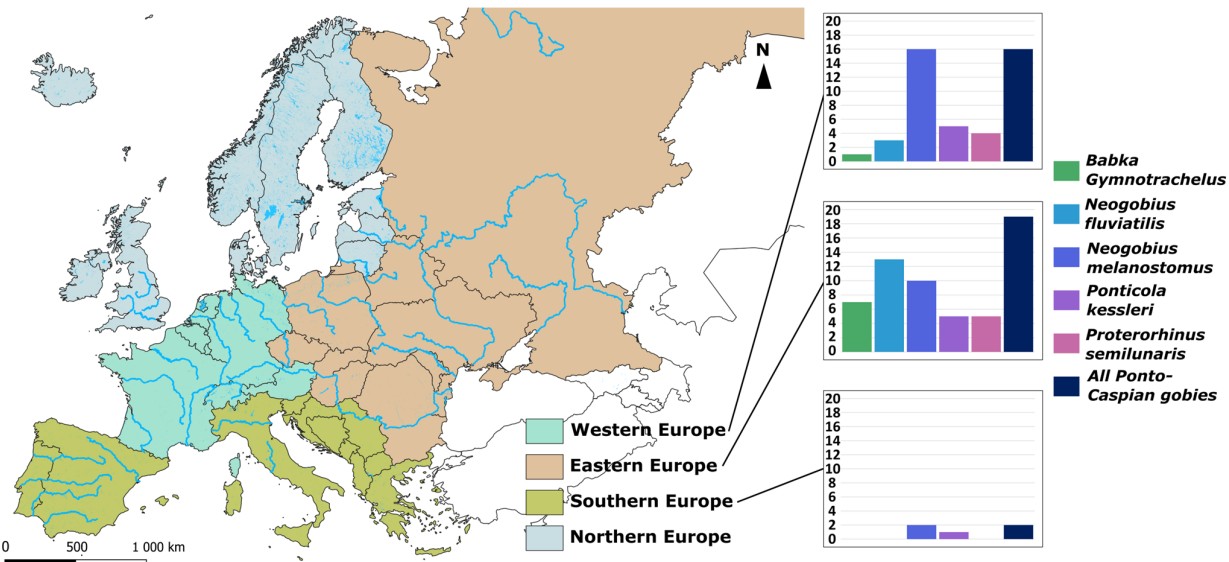

**Figure 8.** Geographical distribution of published studies concerning parasites of Ponto-Caspian Gobiidae in Europe.

A few other species were concerned by spillback effect after the introduction of new Ponto-Caspian Gobiidae hosts: the Digenea *A. gracilis*, *Holostephanus cobitidis* Opravilova, 1968, and *Holostephanus luehei* Szidat, 1936, were recorded in these hosts in several countries in which they are native, and showed a reinforcement of their population in Poland and Russia [156,160]. A third parasite of the genus *Holostephanus*, *Holostephanus dubinini* Vojtek & Vojtkova, 1968 was found in *N. fluviatilis* in the lower Volga in Russia [156]. The Nematoda *Raphidascaris acus* (Bloch, 1779) showed an increase in prevalence in Germany [163]. Some other native parasite species were acquired in newly invaded habitat, without these new acquisition leading to any report of consequences for native fish species, e.g., *Acanthocephalus lucii* (Müller, 1776), was reported from *N. melanostomus* and *P. kessleri* in Austria and Germany [132,157], *P. kessleri* and *K. caucasica* hosted *Camallanus truncatus* (Rudolphi, 1814), in Slovakia and Hungary [40,131] and metacercariae of *Diplostomum* were recovered from all main Ponto-Caspian Gobiidae species from numerous countries in western and eastern Europe, e.g., [131,157,159,160].

6.1.2. Acquisition of Previously Introduced Parasites

Not only can non-native hosts acquire native parasites species, they can also become new hosts to previously introduced non-native parasites. The example of *A. crassus* has already been discussed in the previous part of this review, but another example of this phenomenon is the case of *P. laevis*, a Ponto-Caspian Acanthocephala introduced to several western and eastern European countries with its intermediate Gammaridae host *Dikerogammarus villosus* (Sowinsky, 1894), which was also found to infect Ponto-Caspian Gobiidae in their non-native range [45,120,132,164,165]. *Pomphorhynchus laevis* was reported to infect native *Barbus barbus* (Linnaeus, 1758) and *Squalius cephalus* (Linnaeus, 1758) in France through

the spillover effect [164], and contribute to the extinction of a native Acanthocephala species, *Pomphorhynchus tereticollis* (Rudolphi, 1809) in Germany [163].

Ondračková et al. [119] showed that the parasites acquired by Ponto-Caspian Gobiidae in their non-native range are mainly immature parasites infecting their fish host at larval or subadult stage. The present review supports this idea as the majority of parasites previously cited were recorded as larval stages.

*6.2. Other Notable Acquisitions of Parasites by Non-Native Fish in Europe: Perccottus glenii*

The Chinese sleeper *P. glenii* was first introduced in western Russia at the beginning of the 20th century. Then, through other independent introduction events associated with commercial fish from the Amur river basin, and its use as live bait by recreational fishermen, this species reached numerous countries in Europe [72]. *Perccottus glenii* acquired native fish parasites during its invasion of Europe. Kvach et al. [166] and attracted attention to the potential spillback that several species, namely the Acanthocephala *Acanthocephalus anguillae* (Müller, 1780) and *A. lucii*, the Nematoda *Spiroxys contortus* (Rudolphi, 1819) and *P. tomentosa*, the Cestoda *Paradilepis scolecina* (Rudolphi, 1819), *Valipora campylancristrota* (Wedl, 1855) and *Ophiotaenia europaea* Odening, 1963, the Monogenea *Gyrodactylus luciopercae* Gusev, 1962 and the Digenea *Metorchis xanthosomus* (Creplin, 1846) and *Schiginella schigini* (Bykhovskaja-Pavlovskaja, 1962), could undergo. *Ophiotaenia europea*, recorded from Germany, Ukraine and Russia, is of particular interest as it shows the possibility of establishing parasitological links between non-native fish and native reptiles, as it is transmitted to freshwater snakes *Natrix tessellata* (Laurenti, 1768) and *Natrix natrix* (Linnaeus, 1758) through fish consumption [35,166–169]. This scenario does not only affect wild fauna, but can also have a notable impact on pets, e.g., *M. xanthosomus* was found several times in the introduced *P. glenii* [74,75,166,169] and in *N. fluviatilis* [133] and is of veterinary importance as it is known to infect dogs [170].

Newly acquired parasites may negatively impact the fitness of a non-native species, as with the case of the infection by *Eustrongylides tubifex* (Nitzsch in Rudolphi, 1819). Mierzejewska et al. (2012) [29] noted a strong negative impact on health and condition with destruction of internal organs, inflammatory lesions and castration by *E. tubifex* on *P. glenii*, and thus considered the infection as a potential selective factor able to moderate the population growth of this non-native fish. This particular parasite could thus bring a competitive advantage to native fish species.

As with Ponto-Caspian Gobiidae, and according to the interactions recorded in this review, it seems that the majority of native parasites acquired by *P. glenii* in its non-native range are larval stages, e.g., larvae of *R. acus* [35,74], *S. contortus* [35,40,166,169], *Streptocara crassicauda* (Creplin, 1829) [74], *Triaenophorus crassus* Forel, 1868 [74], *Diplostomum chromatophorum* (Brown, 1931) [35,171] and *Opisthioglyphe ranae* (Frölich, 1791) [35,40]. This observation could thus well be a general trend in non-native hosts/acquired parasites systems, complementary to the accepted idea that these systems include mainly generalist parasites.

## 7. Fish Introduction and Zoonosis

Non-native fish species can be used as hosts by zoonotic parasites and thus be a cause of concern with regard to human health. Fish-borne zoonotic diseases are caused by the consumption of live, raw (e.g., sushi, sashimi and ceviche), smoked, marinated, lightly or inadequately cooked fish [172–175], which serve as intermediate or paratenic host to the parasite. A variety of parasites are concerned with the most focused on being Nematoda, Digenea and Cestoda [172,175]. Well-known nematodiases are anisakidiasis (genus *Anisakis*, *Pseudoterranova* and *Contracaecum*), dioctophymiasis (e.g., genus *Dioctophyme* and *Eustrongylides*) and gnathostomiasis (genus *Gnathostoma*) [172,173]. Zoonotic Digenea comprise both liver flukes such as *Clonorchis sinensis* (Cobbold, 1875) and several species of *Opisthorchis*, and intestinal flukes (e.g., several species of *Metagonimus*, *Heterophyes* and *Hap-*

*lorchis*) [172,175]. Finally, zoonotic Cestoda cause diphyllobothridasis (genus *Diphyllobothrium* and *Dibothriocephalus*) and ligulosis (*Ligula intestinalis* (Linnaeus, 1758)) [172,175].

### 7.1. Eustrongylides

*Eustrongylides* is a genus of parasitic cosmopolitan Nematoda [176–179]. These parasites are responsible for rare human infections, notably in North America and South Sudan; they cause severe abdominal pain and sometimes require surgery to be removed [180–182]. Their life cycle is complex: they use a first intermediate oligochaete host, a second intermediate fish host and a final piscivorous bird host [183]. In Europe, larval stage of *Eustrongylides excisus* Jägerskiöld, 1909, *E. tubifex*, *Eustrongylides mergorum* (Rudolphi, 1809) and unidentified *Eustrongylides* were reported from several non-native fish species: the Ponto-Caspian gobies *B. gymnotrachelus*, *N. fluviatilis*, *N. melanostomus* and *P. semilunaris* [119,122,131,159,160], the East Asian *P. glenii* [29,35,74,169,184,185], and from two salmonids: *S. trutta* introduced into a reservoir in England and stocked *O. mykiss* in Scotland [186,187] (Table 3).

**Table 3.** Host-locality list for zoonotic *Eustrongylides*.

| Parasite Species | Host Species | Locality |
|---|---|---|
| *Eustrongylides excisus* | *Babka gymnotrachelus* | Poland [159,160] |
| | *Neogobius fluviatilis* | Poland [159], Ukraine [159] |
| | *Neogobius melanostomus* | Austria [122,131] |
| | *Ponticola kessleri* | Slovakia [74] |
| | *Perccottus glenii* | Poland [29] |
| *Eustrongylides mergorum* | *Perccottus glenii* | Russia [35] |
| *Eustrongylides tubifex* | *Babka gymnotrachelus* | Poland [160] |
| | *Perccottus glenii* | Poland [29], Ukraine [169] |
| *Eustrongylides* sp. | *Babka gymnotrachelus* | Poland [119] |
| | *Neogobius fluviatilis* | Poland [160] |
| | *Neogobius melanostomus* | Czech Republic [119] |
| | *Proterorhinus semilunaris* | Poland [160] |
| | *Perccottus glenii* | Serbia [184,185] |
| | *Oncorhynchus mykiss* | United Kingdom [186] |
| | *Salmo trutta* | United Kingdom [187] |

Public health concerns may arise as these parasites were recovered from species such as *O. mykiss* and *S. trutta*, which are consumed and prized by anglers. Moreover, their reported presence in numerous non-native fish hosts could lead to a spillback effect, with the risk of increasing their prevalence and/or abundance in fish hosts already present in the habitat. This perspective is concerning, as *Eustrongylides* Nematoda are known to occur in Europe in fish caught in recreational fishing, such as *M. salmoides* and *P. fluviatilis* [177,188,189].

### 7.2. Anisakids: Contracaecum and Anisakis

Anisakid Nematoda are known to accidentally infect human through the consumption of raw or undercooked fish and cause gastro-intestinal pain, vomiting, diarrhea, nausea and allergic reaction [172,190–193]. The majority of cases are attributed to the genus *Anisakis* and *Pseudoterranova*, but *Contracaecum* larvae are also reported. Human cases of *Contracaecum* infections were reported from Australia, Japan and Germany [192]. These parasites have a complex life cycle with a marine mammal or piscivorous bird final host, invertebrate intermediate host [194]. They also use a broad range of fish as paratenic hosts [192,195]. In the present literature review, larval stage of *Contracaecum ovale* (Linstow, 1907) Baylis, 1920, *Contracaecum rudolphii* Hartwich, 1964 and *Contracaecum* sp. were reported from four non-native fish: the Ponto-Caspian gobiids *N. melanostomus* and *N. fluviatilis*, the Centrarchidae *L. gibbosus* and the Leuciscid *C. nasus* (Table 4). Only one record of *Anisakis* Nematoda could be retrieved from stocked *S. trutta* and *O. mykiss* in the United Kingdom [196].

**Table 4.** Host-locality list for zoonotic *Contracaecum*.

| Parasite Species | Host Species | Locality |
|---|---|---|
| *Contracaecum ovale* | *Lepomis gibbosus* | Germany [21,49] |
| *Contracaecum rudolphii* | *Neogobius melanostomus* | Czech Republic [113] |
| *Contracaecum* sp. | *Lepomis gibbosus*<br>*Neogobius fluviatilis*<br>*Chondrostoma nasus* | Bulgaria [48], United Kingdom [34], Poland [197]<br>Slovakia [133]<br>France [51] |

Attention should be paid to the presence of these zoonotic parasites in non-native fish hosts, as the new acquisition of paratenic hosts could result in a spillback to native species [48] and thus increase the abundance of parasites and the prevalence of infection in invertebrate, fish, and bird hosts in invaded habitats. An increased risk of transmission to human through an improperly cooked fish host should then not be excluded.

### 7.3. Clinostomum complanatum

Human infections by *Clinostomum* (Digenea, Clinostomidae) are rare but there are records of laryngitis caused by *Clinostomum complanatum* (Rudolphi, 1814) in Asia [198,199]. These parasites have a complex life cycle with a first intermediate snail host, a second intermediate fish host and a final piscivorous bird host. *Clinostomum complanatum* was reported from the East Asian *P. glenii* in Hungary [40]. This species is already known to occur in Europe in consumed gamefish *P. fluviatilis* and *B. barbus* [200,201]. Attention should be paid to the occurrence of this parasite, which gained an additional host with the invasion of *P. glenii* in Europe.

### 7.4. Metagonimus yokogawai

The Heterophyidae family (Digenea) are intestinal flukes of birds and mammals and contain numerous species reported from humans [172]. This group has been increasingly recognized since the 1990s. Among heterophyids, *Metagonimus yokogawai* (Katsurada, 1912) is considered as one of the most important species, and human cases were recorded from Asia, Middle East and at least two countries in Europe (Russia and Spain) [172,202]. This species is a parasite of the small intestine, where it causes inflammatory reactions; heavy infections can result in abdominal cramps, malabsorption and weight loss [174,175,203]. This species shows rather broad host specificity [172] and was reported in 2006 from three Ponto-Caspian gobies (*N. fluviatilis*, *N. melanostomus*, *P. kessleri*) in Hungary [165]. With the successful invasion of these newly acquired hosts in Europe, a possible spillback effect leading to an increased infection in native fish hosts cannot be ruled out. This parasite is known to occur in Serbia, in several consumed fish species such as *Sander lucioperca* (Linnaeus, 1758), *A. alburnus* and *Scardinius erythrophthalmus* (Linnaeus, 1758) [204]. Even if there is no habitual human consumption of raw fish in this region, uncooked fish are frequently used as a supplementary food source in widespread pig farming [204] and could thus be of veterinary importance.

### 7.5. Dibothriocephalus dendriticus (Syn. Diphyllobothrium dendriticum)

Diphyllobothridasis is reported to be the most frequent fish-borne Cestoda infection in humans, which can be infected by at least 13 species of *Diphyllobothrium* [172]. They appear to have a broad intermediate and final host specificity [172]. As Chai et al. [172] pointed out, there may be a significant risk of spreading diphyllobothriids through the stocking of imported fish because they can act as intermediate hosts in the event of egg contamination. The bibliographical analysis enabled us to retrieve records of one species of *Dibothriocephalus*. *Dibothriocephalus dendriticus* was reported in *G. aculeatus* from Norway (small-scale introduction between two Norwegian lakes) [56] and from stocked *S. trutta* and *O. mykiss* in Scotland and in a reservoir in Essex (UK) [186,196]. Human cases attributed to this species have a circumpolar distribution, with most cases reported from the Lake

Baikal region [172,205]. *Dibothriocephalus dendriticus* is the fourth most frequent causative agent of diphyllobothridasis [205]. As the only recent record of this parasite found in the present bibliographical study is from *G. aculeatus* transferred from one Norwegian lake to another (small-scale introduction) [56], it is unlikely that fish introductions play a major role in human incidence of diphyllobothridasis in Europe.

## 8. Conclusions

According to the bibliographical review conducted here, the introduced fish species that have attracted the most attention in Europe with regard to their parasite communities were the Ponto-Caspian Gobiidae, the North American Centrarchidae and the Asian *P. glenii*. With regard to parasites, *A. crassus* was the most focused on, likely because of its pathogenicity towards such a threatened resource as the European eel. Freshwater fish introductions in Europe offered various examples of known mechanisms such as spillover and spillback. No occurrence of a dilution effect was recorded, but this does not imply the absence of this mechanism in Europe. As already evidenced before, the ability to infect a wide range of intermediate and/or paratenic hosts can be a key factor in the success of a parasite in its non-native range. This was particularly striking in the example of the successful *A. crassus*, which was reported from 35 paratenic host species belonging to 12 families. It is also an example of how non-native fish can facilitate the spread of native and previously introduced parasites. Concerning parasites with complex life cycles, those infecting their fish host at an immature (larval/sub adult) stage tend to be the most successful in using non-native fish hosts. It would be of interest to explore recorded fish host–parasite systems in the context of species introductions in other parts of the world to obtain some insight on whether this observation can be generalized. Non-native fish introductions in Europe may influence zoonotic parasites dynamics, mainly through a spillback effect, as recorded interactions primarily involved zoonotic parasites already present in Europe, but which gained additional non-native hosts. For this reason, the novel acquisition of non-native fish hosts for the zoonotic parasites present in Europe should be taken into account. In a context of growing popularity of dishes including raw or lightly marinated fish such as sushi, sashimi and ceviche, this issue is worthy of greater attention. A concern arising from the present study is the very low number of publications, including insular environments in the study of non-native freshwater fish/parasites. Yet such environments are particularly vulnerable to species introduction and should be more widely focused on in the near future.

**Author Contributions:** Conceived and designed the study, A.E., J.F. and Y.Q.; performed and discussed the work and edited the manuscript, A.E., J.F. and Y.Q. All authors have read and agreed to the published version of the manuscript.

**Funding:** The present study was partially funded as a doctoral fellowship at the University of Corsica Pasquale Paoli and the Culletivittà di Corsica granted to A.E. This research is part of the GERHYCO interdisciplinary project dedicated to water management, ecology, and hydro-ecosystem services in an insular context, and it was financially supported by the Culletivittà di Corsica.

**Institutional Review Board Statement:** Not applicable.

**Informed Consent Statement:** Not applicable.

**Data Availability Statement:** The authors confirm that the data supporting the findings of this study are available within the article and its supplementary material. Raw data that support the findings of this study are available from the corresponding author, upon reasonable request.

**Conflicts of Interest:** The authors declare no conflict of interest.

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
