# Peer review of "Parasitic Helminths and Freshwater Fish Introduction in Europe: A Systematic Review of Dynamic Interactions"

_fishes, doi:10.3390/fishes8090450_

Round 1

Reviewer 1 Report

Very interesting and good prepared article. I have just several small comments due to corrections might be provided. Also, some articles are lost in the review and night be added into analysis in my opinion.

Line 29: space between 'scales:inter-continental'

Line 59: 'Schyzocotyle acheilognathi (Yamaguti, 1934) Brabec, Waeschenbach, Scholz, Littlewood & Kuchta, 2015'. - here and then: the authors of taxonomic revisions are sometime noted in the text, but sometime not. Because the article is not dedicated to taxonomy, this is not important, and enough to use only the authors of the species, e.g. 'Schyzocotyle acheilognathi (Yamaguti, 1934)'. This might be standartised throughout the text.

Line 200: 'Centrarchidaeae' might be 'Centrarchidae' - please correct this throughout the text.

Lines 237-238: 'the Cestoda Nippotaenia mogurndae Yamaguti & Miyata, 1940 and Nippotaenia perccotti (Akhmerov, 1941) Akhmerov, 1960' - this is the same species. See explanation in: Scholz, T., Brabec, J. & Kuchta, R. 2017. Nippotaeniidea Yamaguti, 1939. In: Caira, J.N. & Jensen, K. (Eds.). Planetary Biodiversity Inventory (2008–2017), Tapeworms from Vertebrate Bowels of the Earth, No. 25. University of Kansas, Natural History Museum, Lawrence, KS, pp. 243–250. Also, the short discussion is in doi:10.1515/biolog-2016-0112 - last one is consists of new data and might be included into analysis. The species might be mentioned as Nippotaenia perccotti (Akhmerov, 1941).

Line 281: 'mostly in eastern Europe (16 articles e.g. [48,61–63]' - one more article is doi:10.1007/s00436-021-07268-8

Line 282: 'western (e.g. [49,64,65])' - one more article is doi:10.1017/S0022149X20000267

Table 1: 'Ancyrocephalidaeae' might be 'Ancyrocephalidae' (two times in the table)

Line 310: 'N. mogurndae and N. perccotti' - same species. Might be 'N. perccotti'

Table 2 (Page 13): For Perccottus glenii the reference 'Slovakia, [72]' is cited. It might be a mistake, because the article is about the parasites of Ponticola kessleri in the Danube River, not about Perccottus glenii. I have no information about Anguillicola in Perccottus.

Line 460: 'river Dniester via the Pripyat-Bug' - might be Dnieper, not Dniester.

Lines 474-475: Lack of the information about the introduced Neogobius fluviatilis in the Lower Volga, see doi:10.1515/biolog-2015-0108

Line 494: One of the first discussion about Bucephalus polymorphus in invasive gobiids is doi:10.1051/kmae/2010034 - might be cited here.

Lines 505-506: Not only in Poland, but also in Russia, Lower Volga (doi:10.1515/biolog-2015-0108). Also, Holostephalus dubinini was found in the invasive monkey goby there.

Table 3: 'Austria [119,119]' - 119 two times. 'Slovakia [72]' - the article is about Ponticola kessleri, and indeed Eustrongylides excisus occurred in Slovakia. Please list Ponticola kessleri in the table.

Reviewer 2 Report

The topic of the manuscript is relevant and interesting. A systematic search for literature has been performed and the resulting references used. Clearly the authors have manually added further references taken from reviews. 

The title should be improved as it appears a bit problematic. One could as an alternative  suggest "Parasitic helminths introduced with freshwater fish in Europe: A review of dynamic interactions".

The abstract mentions a few host-parasite systems although numerous are mentioned in the proper manuscript text. One should consider if the abstract should highlighting the general outlines and not focusing on a few systems.

The authors have decided to ignore that Anguillicola was transferred by Moravec and co-authors to the genus Anguillicoloides decades ago. Maybe they should a comment on this decision.

It is surprising that the co-introduction of eg. Dactylogyrus lamellatus with grasscarp is not touched upon. The same goes for the introduction of Gyrodactylus salaris with Swedish Baltic salmon to Norway in the 1970s. In the light of the devastating effect it had in Norway I suggest to include this topic as well.

Generally well written but a final brush-up after revision can be conducted.
